# Revolutionizing Therapeutic Delivery with Microneedle Technology for Tumor Treatment

**DOI:** 10.3390/pharmaceutics15010014

**Published:** 2022-12-21

**Authors:** Vaibhavi Meghraj Desai, Sakshi Priya, Srividya Gorantla, Gautam Singhvi

**Affiliations:** Industrial Research Laboratory, Department of Pharmacy, Birla Institute of Technology and Science (BITS)—Pilani, Pilani Campus, Pilani 333031, India

**Keywords:** microneedles, tumor therapy, tumor diagnosis, theranostic, chemotherapy

## Abstract

The tumor is an uncontrolled growth of tissue that can be localized (benign) or possesses the capability of metastasis (malignant). The conventional methods of tumor diagnosis, such as acupuncture, endoscopy, and histopathology, and treatment methods, such as injections, chemotherapy, surgery, and radiotherapy, are invasive, expensive, and pose severe safety and management issues for the patients. Microneedle technology is a recently developed approach for active transdermal drug delivery. It is minimally invasive, self-administrable, bypasses the first-pass effect, and effectively delivers chemotherapeutics and drugs at low doses, thus, overcoming the drawbacks of conventional delivery systems. This review provides an idea of the types, materials utilized in the fabrication, and techniques used for the preparation of microneedles (MNs), as well as their application in tumor diagnosis and treatment. Additionally, emphasis is given to the case studies related to MNs-assisted tumor therapy, such as photothermal therapy, gene therapy, photodynamic therapy, chemotherapy, immunotherapy, and various combination therapies. MNs also serve as a tool for diagnosis by the bio-sampling of blood and interstitial skin fluid, as well as biosensing various cancer biomarkers. The combined therapy and diagnostics provide theranostic MNs for enhanced and personalized tumor therapy. The limitations and prospects of MNs development are also discussed.

## 1. Introduction

As per the American Cancer Society, cancer is defined as a set of diseases categorized by unrestrained proliferation and propagation of abnormal cells that left untreated may lead to fatality. The uncontrolled growth of new tissue is termed neoplasia and results in tumor formation. A tumor can be localized (benign) or possesses the capability of metastasis (malignant tumors) [1]. As per the latest estimations by GLOBOCAN2020 and WHO, there were 19.3 million cases of cancer and 10 million deaths worldwide for the year 2020. GLOBOCAN2020 also predicts that by the year 2040, there will be a rise in cancer cases to 30.2 million [2,3].

The existing methods of chemotherapy (injections, surgery, chemotherapy, radiotherapy) and diagnosis (acupuncture, endoscopy, and histopathology) pose severe safety and management issues for the patients and are also very expensive [4]. The oral delivery of anticancer agents is associated with the first-pass effect and low bioavailability problems [5]. To avoid these issues, chemotherapeutic agents are given by the IV route, as it provides advantages such as higher drug plasma levels, quicker onset of action, and prevents gastrointestinal side effects; however, this route is associated with a risk of infection, thrombus formation, pain, and hypersensitivity, and its administration requires trained personnel [6]. Even novel therapies such as nanomedicines cause sub-optimal efficacies. They are encumbered by their low capacity to accrue within the tumor tissue, i.e., no more than 0.7% of the dose that is administered reaches the tumor site. Moreover, the synthesis process, scale-up, and reproducibility of the therapeutics outcome are complicated due to high complexity [1,7,8]. The transdermal delivery system is being explored as an alternative route to oral and IV routes to deliver chemotherapeutic agents to the tumor site, as it is relatively less invasive and prevents systemic side effects and high first-pass metabolism [6]. Conventional transdermal delivery systems such as patches and ointments show limited bioavailability due to the major barrier of the skin, the stratum corneum [9]. Moreover, the stratum corneum only allows the diffusion of drugs with a low molecular weight (<600 Da). Therefore, many strategies are employed to enhance the permeation of chemotherapeutic agents via the stratum corneum, such as electroporation, sonophoresis, iontophoresis, needleless jet injector, thermal ablation, and microneedles (MNs) array [6]. MNs are a novel transdermal delivery technology with unique micron-sized needles which easily penetrate the stratum corneum without coming into contact with painful nerves below the dermis region, and they have high efficiency, painless penetration, and sufficient convenience [9].

The objective of this review is to give a broad overview of MNs-based tumor diagnosis, therapy, and recent advancements in tumor theranostics. The manuscript comprises the research outcomes of case studies on tumor therapy, including chemotherapy, photothermal therapy (PTT), photodynamic therapy (PDT), gene therapy, immunotherapy, and combination therapies, as well as discussion on tumor diagnosis, including MN-assisted sampling devices and biosensors. This review also highlights the upcoming MNs-based theranostic approach for a combined tumor diagnosis and therapy along with clinical status, and current perspectives for understanding the significance of MNs in cancer treatment.

## 2. Microneedles in Cancer Therapy and Diagnosis

MNs technology is a form of active transdermal drug delivery and could be used as a substitute for conventional syringe injections. The direct injections of therapeutic agents into the tumor can cause leakage of the drugs to the surrounding healthy tissues, thereby decreasing safety and effectiveness. The basic mechanism of MNs-assisted drug delivery involves the disruption of skin layers by forming micron-sized pathways to directly deliver the drug to the epidermal or upper dermis region [10]; thus, the drug enters systemic circulation without obstructions from the stratum corneum barrier. The MNs array penetrates the stratum corneum, delivers the drug with a minimally invasive action, and provides drug diffusion into the tumor areas [11]. The MNs are available in various shapes and are sized with a height ranging from 150 to 1500 µm and a width ranging from 50 to 250 µm [12,13,14]. MNs have a synergistic combination of the transdermal patch with hypodermic needles, wherein a drug is directly delivered via the micro-channels to the site of tumor, especially in breast, prostate, cervical, and skin tumors. MNs provide controlled site-specific delivery, thereby preventing the destruction of healthy cells [5].

MNs, such as solid MNs, coated MNs, hollow MNs, dissolving MNs, and hydrogel-forming MNs, are explored for therapeutic delivery and the diagnosis of disease conditions. Each type has its own set of attributes: solid MNs are useful in generating pores in the stratum corneum as pre-treatment, but there is also medication-coated MNs, dissolvable MNs for sustained release, or hollow MNs filled with drug solution. All types are employed in cancer diagnosis and therapeutics [13]. The different kinds of MNs and their approaches for transdermal delivery are shown in Figure 1.

A variety of biomaterials, such as silicon, biodegradable and non-biodegradable polymers, metals, carbohydrates, and ceramics, are used to fabricate MNs for specific applications [13,15]. Various fabrication techniques for manufacturing theranostic MNs involve cutting, etching, photolithography, micro moldings, and 3D printing. Cutting is a process where computer-aided design software uses a laser to cut solid MNs from a stainless steel sheet into the desired shape and size. Etching involves the utilization of a powerful liquid acid or a gas to cut layers from the MNs surface to obtain the required dimensions. Photolithography utilizes light sources such as UV radiation to form thin, minutely patterned films by the spin-coating of a liquid photosensitive polymer on a substrate, which on hardening is used to replicate the substrate design. Micro molding produces MNs of silicon by pouring the hot polymer into micro- or nano-sized molds, followed by centrifugation, drying, and cooling. Three-dimensional printing is a newer technique that enables MNs fabrication in a single step, as well as specific customizations. This technique also utilizes a computer-aided design model that precisely place the material in a layer-by-layer pattern to produce MNs with the desired shape and geometry [13,14]. The general characteristics of MNs are shown in Figure 2.

The different types of MNs, along with their characteristics, advantages, and disadvantages, are summarized in Table 1 [9,10,11,15,16,17,18,19,20].

## 3. Microneedles-Assisted Tumor Therapy

To achieve effective topical tumor treatment, MNs are incorporated with many therapies such as chemotherapy, gene therapy, PTT, immunotherapy, and PDT [9]. Figure 3 summarizes the recent developments in MNs-assisted tumor therapy, diagnosis, and theranostics.

### 3.1. Chemotherapy

MNs-assisted chemotherapy enables the topical, localized, and minimally invasive delivery of chemotherapeutic agents with reduced side effects [9]. Hence, MNs-mediated chemotherapy has gained a lot of attention and interest.

Cisplatin is a first-line chemotherapeutics agent; however, its application is rare because of toxicity in systemic circulation and severe side effects. For safe and effective cancer therapy, Lan et al. developed an MNs-mediated transdermal delivery of lipid-coated cisplatin nanoparticles (LCC-NPs). The pH-responsive tumor-targeting lipid NPs encapsulated cisplatin with a high loading rate of 80%, and this led to the enhancement of drug solubility and improvement in its in vitro antitumor potential. The model used for the in vivo study was a xenograft tumor animal model, which showed that MNs-mediated delivery had an apoptotic index of 58.6%, where it caused significant apoptosis and increased cytotoxicity in tumor cells, thereby considerably reducing tumor volume and weight (Figure 4). The in vivo proliferation study showed only 20% proliferation in the MNs-based formulation group, whereas the proliferation rate was 60% in the group that received the formulation systemically. The technique was bio-safe, as it showed the absence of serum platinum, hepatotoxicity, nephrotoxicity, and pulmonary toxicity [21].

Microneedles are also used in the case of ocular melanoma; however, depending on the nature of the cornea, conventional polymeric microneedles will be relatively larger. Kim and coworkers fabricated miniaturized porous silicon (p-Si) bioresorbable MNs covalently associated with the drug at doses comparable to conventional polymeric MNs. These needles remain in the tissue and gradually degrade, enabling a sustained drug release. In in vivo studies on B16F10 murine melanoma cells using a doxycycline (DOX) drug incorporated with p-Si MNs, significant inhibition of melanoma was observed using DOX-loaded p-Si needles, whereas by syringe the same dose was administered and failed to inhibit the growth of the tumor. For ~24 h, more than 80% of the covalently linked DOX was released. This was significantly longer than the control specimens with the physically-bonded DOX (~8 h) and conventional polymeric MNs (typically, 15 min^−2^ h) [22]. Further, to enhance the efficacy of MNs-assisted chemotherapy, recently, combination therapy with PTT, PDT, immunotherapy, and gene therapy have also been investigated and found to have improved therapeutic benefits compared to monotherapy.

### 3.2. Photothermal Therapy

MNs-assisted PTT acts on the tumor cells by causing thermal ablation; this is due to hyperthermia generated from light energy strongly absorbed by photothermal agents such as nanomaterials of carbon, gold, silver, and nanocrystals of germanium. This method utilizes MNs with near infrared radiation (NIR) laser radiation (a negligibly invasive local treatment) that kills tumor cells and triggers drug release from MNs [23].

To obtain an efficient PTT through MNs, it is vital to optimize the mechanical strength to facilitate better penetration and quick dissolution after administration. Tian et al. incorporated PTT with dissolving microneedles (DMNs) fabricated from graphene and poly (acrylic acid) (PAA). They combined the property of mechanical strength from graphene and prompt drug delivery owing to good water solubility and the better biocompatibility of PAA. The prepared MNs showed sound photothermal effects under NIR and could also deliver the drug directly to tumor tissues and generate heat simultaneously [24].

PTT with aggregation-induced emission luminogen (AIEgen) is effective for the superficial treatment of tumors, as AIEgens have better photothermal and photostability conversion proficiency; however, their systemic delivery is difficult due to solubility and biodistribution-related problems. To address this, Wei and colleagues developed a DMNs system incorporated with AIEgen (NIR950) for the treatment of malignant tumor melanoma in skin (Figure 5). The drug solubility was enhanced by NIR950-loaded polymeric micelles (NIR950@PMs), which were then incorporated with DMNs. In an in vivo study on subcutaneous B16 tumor-bearing C57 mice, it was seen that the temperature at the tumor site of the MNs-treated group promptly increased compared to the IV injection group, even though the dosage administered to the IV injection group was 12.6 times that of the MNs group. This therapy showed several advantages, such as ultra-photostability and applications in theranostics; greater accretion of photothermal agent at the site of tumor; better permeation; enhanced comfort; and painless administration [25].

Another application of MNs-assisted PTT for superficial tumor treatment was made by Lin et al. by constructing two-dimensional (2D) MXene (nanosheets of Nb_2_C) and a dissolvable polyvinylpyrrolidone (PVP) MNs delivery system (PVP-Nb_2_C MNs). The nanosheets served as photothermal agents, and PVP acted as a dissolvable matrix system and provided higher strength for MNs penetration. This study was conducted on a BALB/c nude mouse with a 4T1 tumor model, where MNs, after insertion and subsequent dissolution, released the nanosheets in the tumor. This led to an increased temperature of about 70 °C and thermal ablation with the aid of NIR laser irradiation (1064 nm) at the second biological window. After two weeks, the tumor growth suppression was observed to be higher than the control group, and an improvement was seen in the survival rate of the mice [26].

### 3.3. Photodynamic Therapy

MNs-assisted PDT involves the encapsulation of non-cytotoxic precursors—such as the mesotetra (N-methyl-4-pyridyl) porphine tetra tosylate (TMP) and Nile red (NR), and phthalocyanine or photosensitizers, such as 5-aminolevulinic acid, aluminum (III) phthalocyanine tetrasulfonate chloride (AlPcS4), and methyl aminolevulinate (MAL)—in MNs, which, when stimulated under laser irradiation with a specific wavelength, produce cytotoxic reactive oxygen species (ROS, singlet oxygen) that induce the death of a target cell [23].

PDT with 5-Aminolevulinic acid (ALA) as a photosensitizer has been used to treat basal cell carcinoma, actinic keratoses, and squamous cell carcinoma. To prevent severe skin photosensitivity, topical local administration is more favorable; however, the transdermal penetration of ALA is restricted because of its zwitterionic character and hydrophilicity [27]. To facilitate the transdermal delivery of ALA, Zhao and coworkers developed a sodium hyaluronate (HA)-fabricated MNs patch and incorporated ALA. The effects of the ALA HA MNs patch were compared with an ALA injection and studied on a subcutaneous tumor formed in BALB/c nude mice. It was observed that ALA HA MNs has a low dose of ALA (0.61 mg) and showed a 97% tumor inhibition rate compared to ALA injection comprising a high dose of ALA (1.65 mg). The recovery of microchannels due to the MNs patch was quicker (within 3 h). This therapy showed improved photodynamic tumor therapy safely and efficiently [27].

One drawback of 5-ALA PDT is poor penetration in the skin and, therefore, the inability to treat deep skin neoplasias, such as nodular basal cell carcinoma. A better substitute is the delivery of NR (a model hydrophobic dye and a pre-formed photosensitizer) that enables the penetration of light into the skin when activated at longer wavelengths. Due to its lipophilic nature and high molecular weight, its topical administration is challenging [28]. Donnelly et al. incorporated MNs fabricated from the aqueous mixture of mucoadhesive copolymer Gantrez^®^ AN- 139 with the hydrophobic dye NR-loaded PLGA NPs to facilitate transdermal penetration. Tissue penetration studies were conducted on excised porcine skin, which showed a high dye concentration (382.63 ng cm^−3^) in the tissue. This strategy enabled an effective topical administration of hydrophobic agents, hence, overcoming the drawbacks of existing therapies [28].

### 3.4. Immunotherapy

Immunotherapy in cancer is a biological therapy that functions by activation of the immune system to suppress tumor progression. The therapeutics agents, such as immunomodulators, oncolytic viruses, antibodies, and adjuvants, are used for activation of the immune system. The utilization of MNs for immunotherapy is highly advantageous, as they enable the rapid, effective, and painless delivery of vaccines. For this purpose, DMNs are generally preferred for the immediate delivery of vaccines [9]. To boost the immunogenicity of the tumor antigen, hepatitis B core protein virus-like particles (OVA-HBc VLPs) were fused with the tumor antigen peptide (OVA_257–264_: SIINFEKL) (Figure 6). Mesoporous silica NPs (MSN) were used as a vaccination adjuvant to improve dendritic cells’ performance (DCs). MSN and OVA-HBc VLPs were encapsulated into MNs (MSN/OVA-HBc@MNs). In vitro MSN/OVA-HBc@MNs dramatically promoted DC maturation and enhanced OVA presentation on DCs. MSN/OVA-HBc@MNs could be utilized as preventative vaccinations to successfully suppress the development of tumors and could effectively elicit an antigen-specific antitumor immune response [29].

To produce a strong immunogenic response in the target cells, Duong et al. devised a strategy of using an array of DMNs cocktails. It was prepared with a bioresorbable polypeptide matrix with nanopolyplex, comprising cationic amphiphilic conjugates with ovalbumin-expressing plasmid OVA (pOVA) and immunostimulant-polyinosinic: polycytidylic acid (poly(I:C)) synthesized in the one-pot synthesis method. The pOVA and poly(I:C) were effectively transferred into the intracellular compartments of macrophages and dendritic cells. This study, conducted on 6-week-old BALB/c female mice, showed a more robust antigen-specific antibody response than traditional vaccines. It demonstrated a remarkable enhancement in the reactivation of memory B cells and enhanced the lung clearance of cancer cells [30].

A common problem with DMNs is that they are often fabricated with polymers which are water-soluble and unable to deliver drugs that have poor solubility in water. To solve such an issue, Park and coworkers devised amphiphilic block copolymer NPs that can incorporate poorly water-soluble drugs in MNs fabricated with water-soluble polymers. This system stimulated the immune system via tumor specific antigens and by the growth suppression of cancer cells using Pluronic F127 (F127) NPs loaded with R837 and coated with cancer cell membranes. The prepared NPs were then loaded into rapidly DMNs. These MNs inhibited the cancer cell growth during prophylactic vaccine therapy and immunotherapy. After immunization using MNs on day 21, in comparison to the control group, the tumor volume retarded eight-fold, which demonstrated the therapeutic effect on tumor growth suppression [31].

MNs have also been amalgamated with controlled drug delivery technologies to augment the effectiveness of cancer vaccines. Kim et al. developed a biodegradable MNs patch to deliver a HA-antigenic peptide conjugate for prophylactic immunotherapy in cancer. The MNs were loaded with cytotoxic T-cell epitope peptide conjugated to HA to enable effective antigen delivery to the skin’s immune system. It exhibited an effective response for more than 24 h post administration and demonstrated statistically significant tumor growth inhibition in a B16 melanoma mice model [32].

### 3.5. Gene Therapy

Gene therapy is performed by inserting the genetic material into the target cell to replace and rectify the faulty genes in the cells. MNs are also used as physical delivery and vector-based technologies (viral or non-viral vector) [9].

The conventional methods for treating melanoma have limited therapeutic efficacy due to high toxicity, limited selectivity, and drug resistance. For melanoma, small interfering RNA (siRNA) therapy is gene therapy and has a relatively lower toxicity and high specificity. However, the nature of siRNA is hydrophilic and has a high molecular weight (~13 kDa) with a negative charge, which hinders the penetration of these agents through the stratum corneum [33]. Pan et al. incorporated MNs for gene therapy to enable pain-free but effective transdermal delivery. Polyethylenimine (PEI, 25 kDa) was applied as a carrier for improvement of the cellular uptake of siRNA. The study conducted on B16F10 melanoma tumor-bearing mice showed that, compared with the untreated animal, the tumor volume and weight declined by ~80% after treatment with a dose of 264 µg of STAT3 siRNA, while the tumor weight and volume reduced by ~50% after treatment with a dose of 132 µg of STAT3 siRNA. STAT3 siRNA delivery through DMNs is a practical technique with the aim of inhibition efficiency and marginal adverse effects for skin melanoma treatment [33].

BRAF gene silencing is the leading therapy for melanoma, even though delivering macromolecular siRNA to the site of the tumor is challenging. Ruan and coworkers developed a siBraf delivery system based upon cell-penetrating peptide octaarginine (R8) nanocomplexes in combination with coated MNs, depending on the siRNA internalization by A375 cells. The study was conducted on a murine melanoma model induced in Balb/c-nude mice. There was ~31% downregulation of BRAF mRNA by R8/siBraf nanocomplexes in vivo, and significant downregulation of BRAF protein expression by coated MNs in vivo (Figure 7). This showed MNs as a promising approach for gene delivery in melanoma therapy [34].

MNs have a limited loading capacity, due to which pDNA transfer is challenging. Moreover, pDNA has a minimal transfection efficacy. Cole et al. combined RALA (a cell-penetrating cationic peptide) and pDNA to produce NPs and enhanced the loading capacity of MNs using the lyophilization technique. This new fabrication method enabled the loading of a seven-fold concentration of NP cargo, as well as57 μg of pDNA and 496 μg of RALA peptide per MN. The efficacy of this formulation was checked in a preclinical cervical cancer model, where it evoked a 1.7-fold higher antigen-specific humoral response than those elicited by intramuscular administration. This led to enhanced gene delivery and significant tumor suppression compared to injection delivery [35].

### 3.6. Combination Therapies

In cancer management, the combination of multiple treatment modalities becomes a safe and effective strategy [36]. A combination of chemo and PTT is alleged to improve the potential and effectiveness of tumor therapy to a stage where eradicating tumors is also possible. MNs consisting of PVP with chitosan and poly (vinyl alcohol) were developed for aiding the DOX and gold-core silica shell (AuMSS) nanorods (Dox@MicroN) delivery to cancer cells [37]. These MNs patches showed good photothermal properties, increasing by 12 °C under NIR light irradiation (808 nm, 1.7 W/cm^−2^ for 5 min). This system was in conjunction with chitosan’s pH sensitivity, providing a layer-dependent controlled drug release. This technique is used for cervical cancer therapy and exhibits a superior cytotoxic effect due to chemotherapy and the PTT combination [37].

In order to use Cu-PDA NPs for PTT and chemodynamic therapy synergistic treatment against cutaneous melanoma, the NPs had to first be prepared before being implanted into MNs. Cu-PDA NPs might be utilized to absorb the energy from NIR irradiation and transform it into heat. On the B16F10 tumor-bearing mouse, the obtained core-shell MNs showed excellent antitumor activities in vivo (Figure 8). The research presented a new path for the creation of MNs-based microdevices for the synergistic treatment of cutaneous melanoma by the transdermal distribution of multimodal nano-formulation with minimal invasiveness [38].

Chemotherapy and PDT combination have also been used to treat superficial tumors, wherein chemotherapeutic agent cisplatin and photosensitizer IR820 were incorporated into MNs for breast cancer therapy. Fu et al. reported this technique, wherein vinylpyrrolidone-vinyl acetate copolymer (PVPVA) was used as the base material for MNs and incorporated with cisplatin and IR820. It was observed that MNs-mediated cisplatin delivery (chemotherapy) and IR820 delivery (PDT) alone showed only a moderate ratio, i.e., 52.0% and 64.0%, of tumor inhibition, respectively, whereas an MNs-mediated combination delivery of cisplatin and IR820 showed a higher inhibition ratio of the tumor of 90.0%. The synergistic therapy-induced intracellular ROS generation, caspase III activation, and the death of the cell led to profound tumor growth suppression, reduced cell proliferation, and augmented apoptosis. Moreover, the synergistic therapy also showed lower toxicity and better biocompatibility [39].

Transdermal delivery would be a better route for immunotherapy for the skin, as it consists of large amounts of resident antigen-presenting cells. In this case, MNs can facilitate efficient transdermal delivery in the tumor cells microenvironment. Lan et al. devised an MNs patch consisting of aPD-1 and cisplatin (CDDP) loaded into tumor-targeted lipid NPs for precise delivery to cancer tissues. An aPD-1/CDDP@NPs caused cell apoptosis of 24.5%, however only 2.79% cell apoptosis was caused by free CDDP. The cell apoptotic index was 61.4% in the aPD-1 MNs group and 73.2% in the aPD-1/CDDP@NP MNs group, considerably greater than 5.4% in the aPD-1 group. The amalgamation of immuno and chemotherapy has shown to be advantageous for cancer therapy [40].

The p53 tumor suppressor gene therapy has been utilized for the treatment of cancer; however, it is insufficient as a monotherapy due to lack of a sustained treatment and lower therapeutic efficacy. An MNs-assisted combination of gene and PTT was developed to enhance treatment efficacy. An MNs patch fabricated with HA incorporated p53 DNA and IR820 as the photosensitizer. In vitro antitumor efficacy was studied in a BALB/c nude mice model with a subcutaneous tumor, using KB cells (human oral epidermoid carcinoma). The tumor volume was 832% that of the initial volume in the untreated group; the “IR820 MNs patch + NIR” group consisted of 248% that of the initial volume; and the “p53 DNA/ IR820 MNs patch without NIR” group consisted of 367% that of the initial volume. However, in the “p53 DNA/IR820 MNs patch + NIR” group, the tumor volume was only 40% of the initial volume and was considerably smaller, suggested that synergistic therapy had the highest efficacy in tumor therapy [41].

Another MNs-assisted combination therapy, depicted in Figure 9, by Chen et al., integrated PDT and immunotherapy via the co-administration of photosensitizer and anti-CTLA4 (Cytotoxic T-lymphocyte-associated antigen 4) antibody. Dextran NPs (pH-sensitive) were utilized to entrap poorly water-soluble zinc phthalocyanine and water soluble aCTLA4 antibodies and were combined with HA-fabricated MNs. An in vivo antitumor immune response studied on female BALB/ c mice demonstrated that the cytotoxic T-lymphocyte count and T lymphocytes count in the group receiving combination therapy was higher (7.58% and 7.04%) than the group receiving only immunotherapy (0.27% and 0.65%) and only PDT (1.32% and 1.28%) and the control group (0.90% and 1.05%). Combination therapy had, therefore, enhanced the efficacy and specificity of the treatment [42].

## 4. Microneedles-Assisted Tumor Diagnosis

MNs for tumor diagnosis can be utilized in two ways: as biosensors that detect various biomarkers associated with a tumor, or as a sampling device for the collection of interstitial fluid, which can help in obtaining information for tumor diagnosis [9].

### 4.1. Microneedles as Biosensors

Nitric oxide (NO) is one of the indicators used to detect cancer progression. The overexpression of the induced nitric oxide synthase (iNOS) enzyme is associated with the early stage of cancer that later promotes angiogenesis for cell growth. Various electrochemical sensors are available for the real-time detection of NO, but the detection of submicromolar concentrations is a challenge. A field-effect transistor (FET) that can detect the submicromolar concentrations of NO utilizes gallium arsenide (GaAS) semiconductor, which is cytotoxic and has a severe risk of carcinogenicity. To address this, D. Keum et al. developed an MNs sensor on the endomicroscope with polycaprolactone (PCL) MNs, and a poly (3,4-ethylene dioxythiophene (PEDOT) coating was developed. They were functionalized with hemin molecules for targeting action, as they have a high affinity to NO for the endomicroscopic imaging and biosensing of colon cancer. On testing the microsensor on a mouse-skin melanoma model, the NO levels observed in melanoma tissue were two times higher than in normal tissue. It provided a dual diagnostic system to identify polyp regions in the colon and detect increased NO production in the cancer-specific areas. This system was an effective tool for the rapid diagnosis of cancer. Figure 10 provides a detailed depiction of the microneedle sensor [43].

Another cancer biomarker is vascular endothelial growth factor (VEGF). Song et al. developed the first complementary metal-oxide-semiconductor (CMOS)-implemented capacitive biosensor that uses MNs (aptamer-based functionalized with receptors for peptide). The factor VEGF on an electrochemical reaction with the peptide receptors on MNs altered the capacitance between two MNs. The biosensor measured both small and large variations in capacitance with a 15 fMrms resolution in the 0.1 to 1000 pM range, therefore, showing high selectivity of VEGF detection. Since the biosensor was MNs-based, the dimensions of the sensor system were considerably decreased in comparison to earlier works [44].

Tyrosine kinase (TYR) is a polyphenol oxidase used for melanin synthesis, and its overexpression and deposition in skin cells cause skin melanoma; hence, it is a melanoma biomarker. TYR screening is, thus, essential for the early diagnosis of melanoma. The existing methods are electrochemical and involve biopsies and are, therefore, invasive. Ciui and coworkers developed a wearable bandage device incorporating MNs biosensors, enabling the non-invasive and rapid detection of melanoma. It consisted of immobilized catechol, which formed benzoquinone detected by amperometry upon reaction with TYR biomarker. The obtained signal from the current was proportional to the level of TYR present [45].

### 4.2. Microneedles as Sampling Tools

Interstitial skin fluid (ISF) is a source for detecting various biomarkers for cancer diagnosis and progression. A swellable MNs patch made from methacrylated hyaluronic acid (MeHA) was developed and then crosslinked through UV irradiation. As MeHA had an excellent water affinity and structural integrity, no residues were left in the skin. MNs extracted 1.4 ± 0.3 mg (≈1.4 ± 0.3 µL) of ISF in 1 min and 2.3 ± 0.4 mg (≈2.3 ± 0.4 µL) of ISF in 10 min from an experiment conducted on mice. The extracted ISF was recovered by centrifugation; therefore, it was an easy-to-use and minimally invasive device [46]. Another fabrication material used to prepare swellable hydrogel MNs was gelatin methacryloyl (GelMA), which was developed by Zhu et al. The properties of the patch could be adjusted by changing the concentration of the polymer and time utilized for crosslinking. In an in vivo study on rats, the patch effectively and quantitatively detected glucose and vancomycin in the extracted ISF [47].

For cancer diagnosis, a potential biomarker was reported to circulate—cell-free nucleic acids (cfNAs); among them, a class of 19–25 nucleotides in length—non-coding RNAs called microRNAs (miRNA)—had great potential as biomarkers for cancer, either individually or in combinations. MNs were developed for the sequence-specific sampling, isolation, and detection of nucleic acid biomarkers from ISF fabricated with an alginate–peptide nucleic acid (PNA) hybrid material. These MNs patches had the capacity of collecting samples of up to 6.5 μL of fluid in 2 min, hence, providing faster sampling kinetics and minimally invasive sample collection compared to reported methods that were either invasive (venous blood) or provided poor quality samples (fingerstick capillary blood) [48].

## 5. Microneedles-Assisted Tumor Theranostics

Theranostic systems amalgamate therapeutics with diagnostics in a single system to enable the real-time monitoring of biomarkers for specific diseases and personalized therapy. This approach was reported by Yuan et al. for MNs-assisted deep brain imaging and in vivo laser ablation to diagnose and treat brain tumors. The ultracompact MNs were prepared by fusing 800 nm optical coherence tomography (OCT) imaging and laser ablation. This device provided histology-like imaging of the brain microstructure, which offered guidance of placement to the image of the ablation probe in the deep brain. In the study conducted on a patient-derived mouse model for the imaging of deep brain cancer, the device demonstrated its potential to detect in vivo lesions in the deep brain with a 1.23 mm imaging depth and 2.4 µm axial resolution in air; minimal invasiveness; quantitative tissue attenuation property (ability to differentiate brain cancer from non-cancer tissues); and high speed. The device was integrated with a 1448 nm continuous-wave laser at 350 mW of power, which performs the function of tissue ablation with an ablation depth of about 600 µm. This approach effectively integrated tissue imaging, tissue ablation, and characterization of the ablation process in vivo and in situ via real-time quantitative monitoring [49].

Another theranostic approach that was reported by Lee et al. for brain tumor diagnosis and treatment via MNs involved the delivery of NPs and high-energy photons to the tumor site using MNs on a bioelectronics system (Figure 11). In this approach, the two forms of MNs used were HA-based bioresorbable microneedles (BMNs) consisting of theranostic NPs and epoxy-based light-guiding microneedles (LMNs) incorporated with the stretchable bioelectronic system. The BMNs dissolved and released the NPs to target brain tumors by locally crossing the BBB, which was challenging in the case of intravenous NP delivery. The LMNs detected the location of the NPs via magnetic resonance imaging (MRI) and generated high-energy photons via light-emitting diodes (LEDs), as well as through embedded alumina (Al_2_O_3_) microparticles, guiding them to the NPs inside tumor for activation. The effectiveness of this treatment was tested on the glioblastoma mouse tumor model, where a time-dependent alteration in the tumor model was assessed via T2-weighted MR imaging. It was observed that the negative control group showed a 520% change in mean tumor volume, and the therapy group showed a −12% change after two weeks. This affected the survival rate where the therapy group had a 42-day median survival time, which was longer than the negative control group by 24 days [50].

Another interesting study by Gittard et al. highlighted quantum dots (QD) applications, which were nanostructured fluorescent semiconductor materials with a diameter of 2–10 nm as theranostic agents, as they had an ability to act as fluorophores and vehicles for drug delivery. The problem with these was that QDs (which were topically applied) showed poor penetration, as they accumulated in the uppermost epidermal region. To facilitate the transdermal delivery of QD, hollow MNs devices were fabricated using a two-photon polymerization process from an acrylic polymer. Therefore, by using MNs, the QD solution was successfully delivered to the dermis and deep epidermis. This enabled the simultaneous multiphoton microscopic imaging of the QD and drug delivery, thus, providing theranostic treatment [51].

The above-mentioned case studies are summarized in Table 2.

## 6. Clinical Status of Microneedles-Based Tumor Therapy

The majority of pre-clinical studies on the application of MNs in tumor therapy were found to be operational in multiple outlooks, but only a few have been successful in human subjects. The clinical trial currently recruiting participants to study the efficacy or safety of DOX MNs for basal cell carcinoma will be conducted in two parts. Part 1 will determine the factors that affect the rate and magnitude of the dissolution of MNs array, such as duration of exposure, age, anatomic location, and the cruciality of proper application technique. Part II will include a randomized study. The safety and efficacy of 50 µg, 100 µg, and 200 µg doses of DOX in a tip-loaded DMNs array will be evaluated via a biopsy of the treated subject [52]. Another ongoing trial on DOX-loaded MNs is for MNs-directed chemo-immunotherapy to locally destroy tumor cells and alter the tumor microenvironment for robust systemic tumor-specific immunity [53]. Information regarding clinical trials for MNs-assisted tumor therapy is discussed in Table 3 [54,55,56,57].

## 7. Patent Status of Microneedles for Therapeutic Delivery and Diagnosis in Tumor

In recent years, there has been great progress in the evolution of MNs as a transdermal drug delivery system [58]. Table 4 summarizes observed patents with respect to the manufacture and applications of various MNs for tumor therapy [58,59,60,61,62,63,64,65]. A major focus was observed to be on tip-coated and biodegradable MNs because of the low dose requirement of chemotherapeutic and immunotherapeutic agents. PDT has recently been widely explored with patents issued as well as clinical trials, as mentioned in the previous section. Patents have also been issued for MNs-based diagnosis as an electronic wearable device for quick diagnosis and review, and as a device for the detection of specific biomarkers. The ongoing research, patents, and clinical trials show that MNs are gaining increasing applications in broad areas—especially in oncology.

## 8. Conclusions and Future Perspectives

MNs are an emerging diagnostic and drug delivery system with unique features such as rapid drug delivery, painless administration, the prevention of injuries or infections due to hypodermic needle-based deliveries, bypass first-pass metabolism, localized therapy, enhanced bioavailability by transdermal administration, and a very compact and easy-to-use system with enhanced patient compliance [67]. These advantages contribute significantly to cancer and tumor therapy and diagnosis, which is challenging by conventional methods. MNs-assisted monotherapy and combination therapies for tumors overcome the majority of drawbacks, such as the effective and localized delivery of chemotherapeutic agents at low doses; site-specific delivery, preventing damage to healthy tissues; and the rapid and effective delivery of NPs, liposomes, and other delivery agents, which are difficult to administer by conventional routes. MNs are also effectively used as diagnostic tools to bio-sample blood or ISF and biosensing various cancer biomarkers, enabling rapid and early detection which can allow the timely initiation of treatment. MNs are also applied as theranostic tools that combine therapy and diagnosis, such as tumor imaging-guided therapies, and are ideal for personalized tumor treatment. Nanocarrier systems with pH-responsive and enzyme-responsive properties were also successfully incorporated with MNs. To cater to the requirements of controlled release, MNs were designed as multi-layered or core shell structures, which shows their practical and wide applicability with innovative designs [9].

Despite all these successful applications and investigations, substantial challenges still need to be addressed before their use in real life. The short-term applications may not cause significant safety-related problems; however, long-term usage, especially in the case of cancer therapy, might be associated with concerns such as skin erythema—with a prompt recovery within 0.5–2 h [68]—and in the case of solid MNs can also induce pain and numbness depending on the length and size of the MNs [69,70]. A balance should be maintained between the flexibility and strength of MNs for effective penetration. The majority of research focuses on achieving higher strength for better penetration; however, flexibility determines its integrity in the skin and impacts drug utilization and therapeutic efficacy; hence, it must be duly considered [9]. For long-term use, the patients must be trained for self-administration, whereas applicators can be used for proper administration in the case of cancer patients who have an extreme weakness. Biodegradable and hydrogel-forming MNs can lead to the high dose delivery of anticancer agents, which is undesirable, and hence, repetitive usage is often restricted [71,72]. Tip-loaded MNs can overcome this issue [27,37]. Controlling the drug loading in MNs is still a challenge due to their inherent preparation techniques. Moreover, the utilization rate during administration is also nonuniform, causing variations in dosing. Therefore, stability in the administration of MNs should be controlled, otherwise it can alter the efficacy [9]. Despite the increasing application of NP-mediated delivery [73], more research is required to achieve assurance of its safety, in vivo biological fate, and its effect on pharmacokinetics to obtain an effective and successful MNs-assisted tumor therapy.

## Figures and Tables

**Figure 1 pharmaceutics-15-00014-f001:**
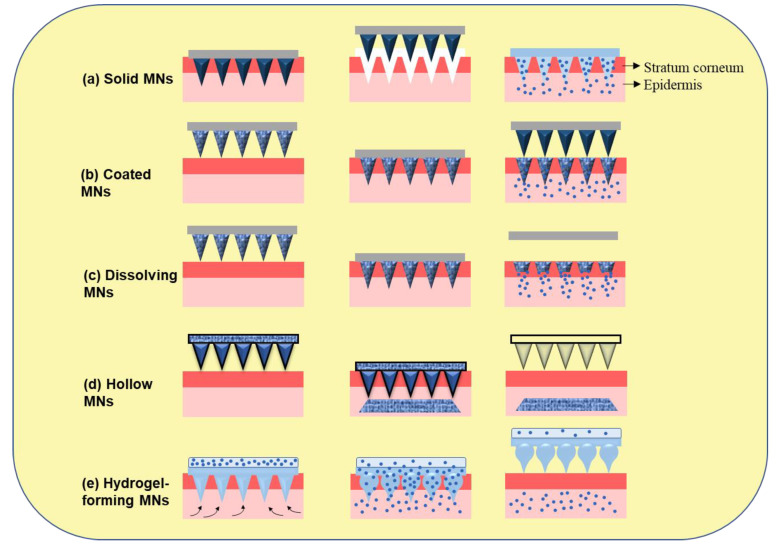
Different types of microneedles and their approaches for transdermal delivery. (**a**) Solid MNs (poke and patch); (**b**) coated MNs (coat and poke); (**c**) dissolving MNs (poke and dissolve); (**d**) hollow MNs (poke and flow); (**e**) hydrogel-forming MNs (poke and release).

**Figure 2 pharmaceutics-15-00014-f002:**
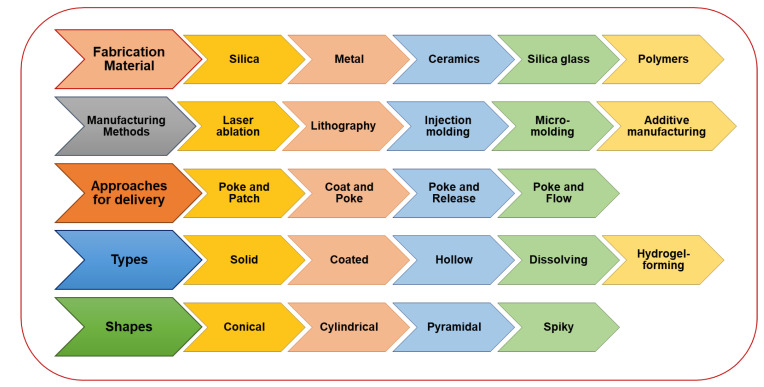
An overview of major aspects of microneedles.

**Figure 3 pharmaceutics-15-00014-f003:**
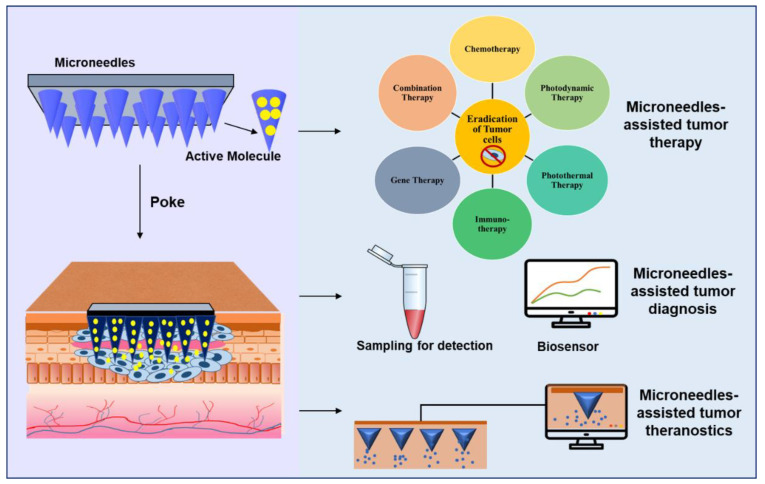
Microneedles-assisted tumor therapy, diagnosis, and theranostics.

**Figure 4 pharmaceutics-15-00014-f004:**
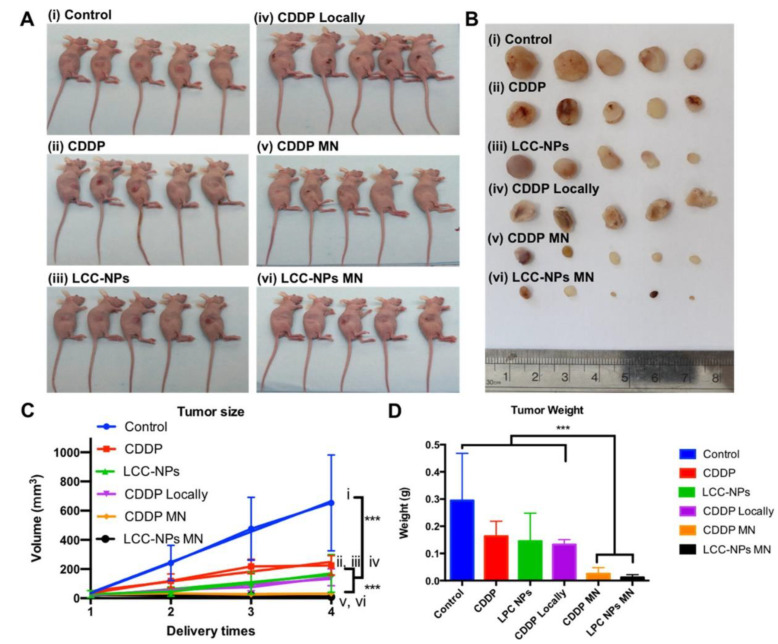
Antitumor efficiency of different groups. (**A**) Photograph of mice before sacrifice and (**B**) tumors harvested in each group: ((i) control, (ii) CDDP, (iii) LCC-NPs, (iv) CDDP locally, (v) CDDP MN, and (vi) LCC-NPs MN); (**C**) tumor volume of each group at 4 time points (1, 2, and 3 represent times of each injection; 4 represents 3 days after the third injection); (**D**) tumor weight of each group: each bar represents mean ± SD (*n* = 5); analysis of variance was calculated using the Mann–Whitney *U* test; *** *p* < 0.05. (Reprinted with permission from Ref. [21]. Copyright 2018 American Chemical Society).

**Figure 5 pharmaceutics-15-00014-f005:**
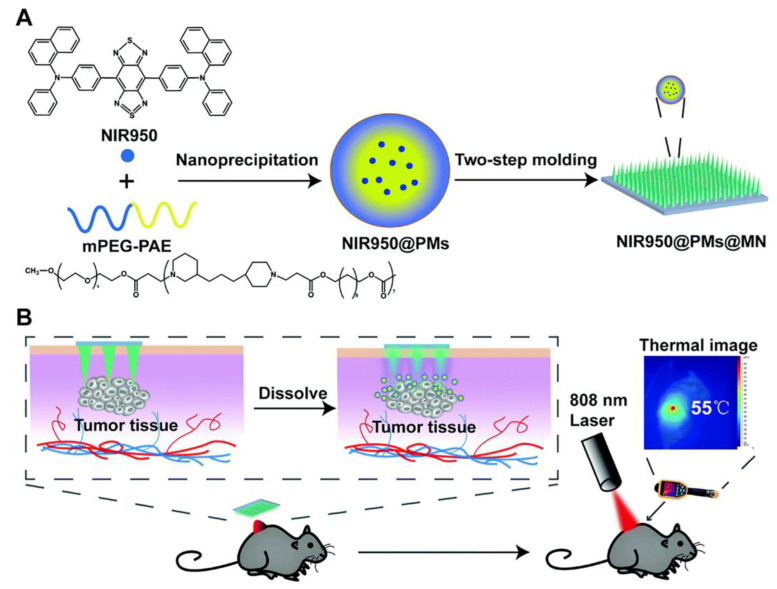
(**A**) Schematic illustration of the fabrication process of NIR950@PMs and NIR950@PMs@MN. (**B**) Schematic illustration of NIR950@PMs@MNapplication process and photothermal effect on B16 tumor-bearing C57 mice. (Reprinted with permission from Ref. [25]. Copyright 2020 Royal Society of Chemistry).

**Figure 6 pharmaceutics-15-00014-f006:**
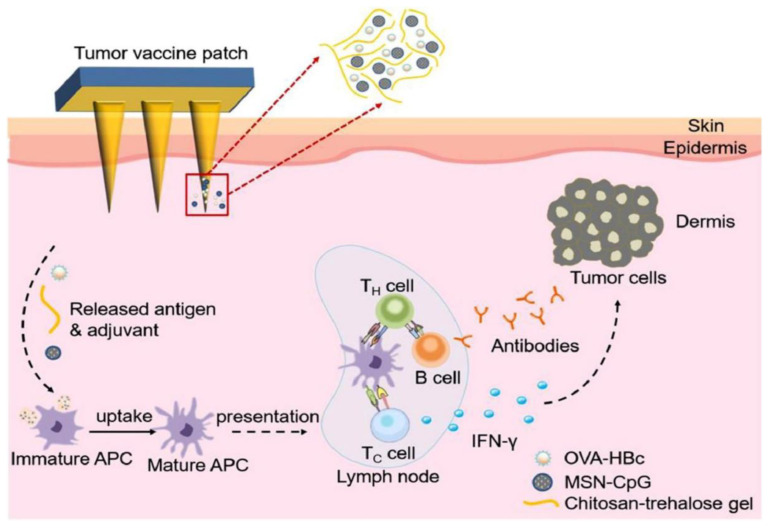
Scheme of MSN/OVA-HBc @MN-mediated antitumor response. (Reprinted with permission from Ref. [29]. Copyright 2021 Elsevier).

**Figure 7 pharmaceutics-15-00014-f007:**
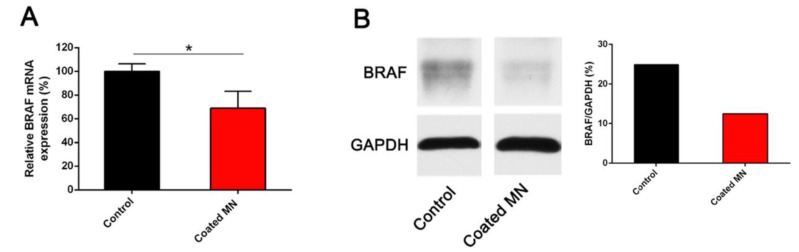
In vivo BRAF gene silencing of R8/siBraf coated MN on melanoma-bearing mice. Down regulation of BRAF mRNA expression (**A**) and BRAF protein expression 9 (**B**). * *p* < 0.05, *n* = 3. (Reprinted with permission from Ref. [34]. Copyright 2018 Elsevier).

**Figure 8 pharmaceutics-15-00014-f008:**
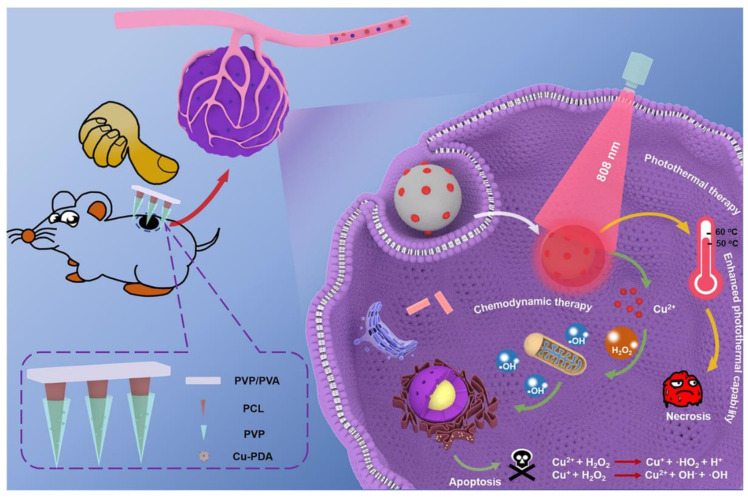
Schematic illustration of Cu-PDA-loaded MNs for photothermal (PTT) and chemodynamic (CDT) synergistic therapy against skin melanoma. (Reprinted with permission from Ref. [38]. Copyright 2021 Elsevier).

**Figure 9 pharmaceutics-15-00014-f009:**
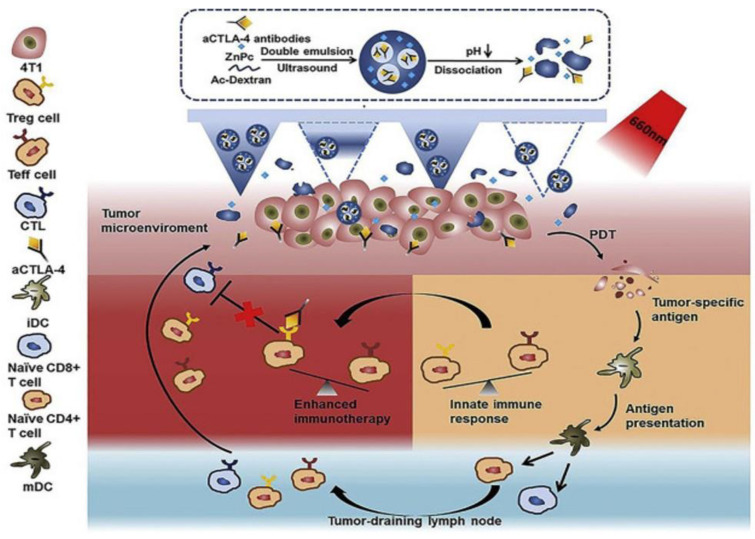
Scheme of the MN-assisted co-delivery system and possible mechanism of antitumor immune responses induced by MN-assisted PDT in combination with immune checkpoint blockade. (Reprinted with permission from Ref. [42]. Copyright 2020 Elsevier).

**Figure 10 pharmaceutics-15-00014-f010:**
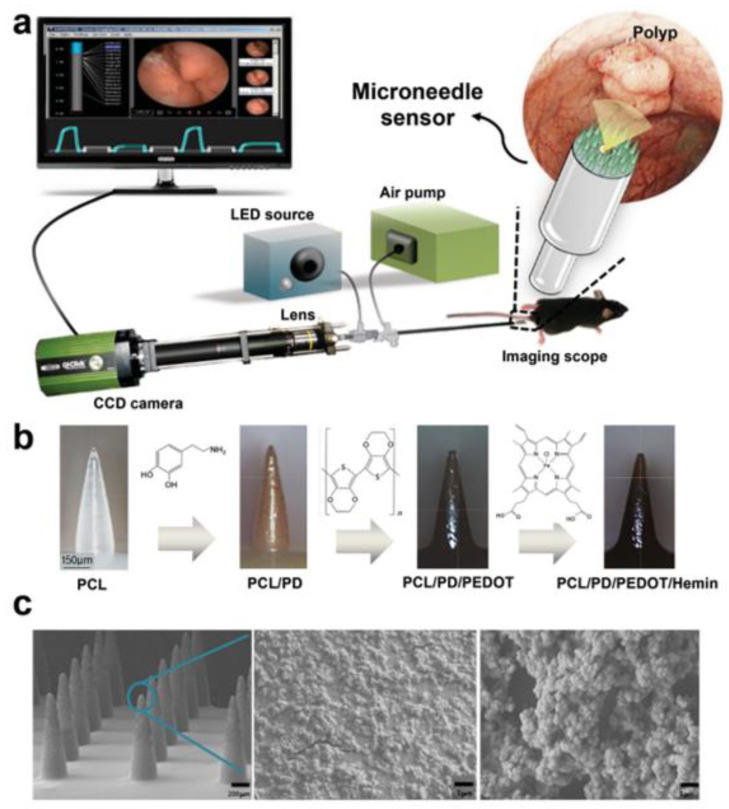
(**a**) Schematic illustration of a microneedle sensor (MS) fabricated at the end of an endomicroscope for both imaging and sensing of cancer. (**b**) Optical microscopic images of MS according to the fabrication process. (**c**) Scanning electron microscopic images of MS (200 µm scale) and the enlarged images of a needle end point (5 µm and 1 µm scale). (Reprinted with permission from Ref. [43]. Copyright 2015 John Wiley and Sons).

**Figure 11 pharmaceutics-15-00014-f011:**
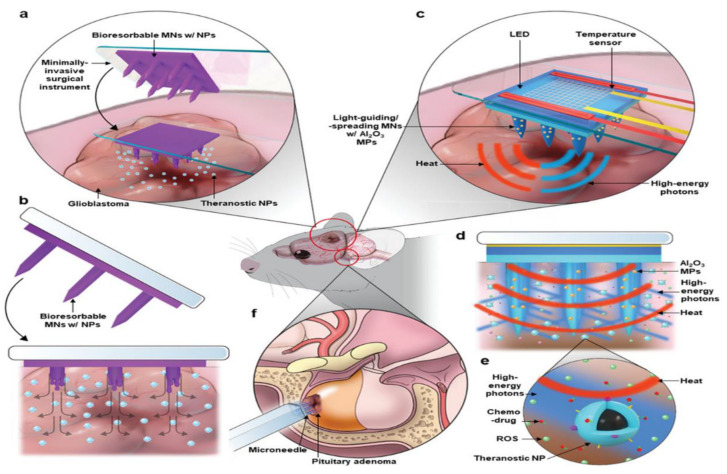
Localized delivery of theranostic NPs and their activation by high-energy photons. (**a**) Localized delivery of theranostic NPs to glioblastoma, bypassing the BBB by using BMNs. (**b**) Biodegradation of the BMNs and release of the NPs inside the target tumor. (**c**) Delivery of the high-energy photons via the LMNs with Al_2_O_3_ microparticles (MPs) and controlled application of the heat by bioelectronics. (**d**) Activation of the NPs by the high-energy photons and accelerated diffusion of the therapeutic agents by controlled thermal actuation. (**e**) Release of the ROS and DOX from the NPs by the irradiation of the high-energy photons. (**f**) Application of the proposed method to the minimally invasive treatment of pituitary adenoma. (Reprinted with permission from Ref. [50]. Copyright 2021 John Wiley and Sons).

**Table 1 pharmaceutics-15-00014-t001:** Characteristics of various types of microneedles and their advantages and disadvantages.

Types of Microneedles	Characteristics	Advantages	Disadvantages
Solid	Poke with patch approach is utilized for skin pre-treatment; Penetration of drug until lower layers of skin due to channel formation; Sharper tip.	Simpler manufacturing; Higher drug loading capacity.	May lead to skin damage or micro incisions over a period of time; Closing of the area is needed to avoid infections.
Hollow	Poke and flow approach; Deposition of filled drug solution in the dermis; Controlled drug release can be achieved.	High amount of drug can be loaded and delivered; Applications in sampling and diagnosis.	Low strength, delicate, and requires utmost care while designing and insertion; Prone to leakage and clogging.
Coated	Coat and poke approach, needle surface coated with drug solution; Minimally invasive and can deliver low quantities of drugs, DNA, and proteins; The size of the needle and thickness of the coat impacts drug loading capacity (which is generally very small).	Faster delivery of the drug as it is present on the surface.	Prone to infection.
Dissolving	Poke and release approach; Formed of biodegradable polymers; Rapid release of macromolecules can be achieved.	Easy to administer; One-step application; Does not require removal.	Complex manufacturing process; Requires more time to dissolve.
Hydrogel-forming	Formed of polymeric composition with “super swelling” properties; Potential to acquire a large quantity of water into their three-dimensional arrangement; Capillary circulation and drug patch is linked via channels formed; Function as a rate-controlling membrane.	Size and shape have flexibility; Simple sterilization process and intact removal from the skin.	Slower rate of drug absorption into the plasma; Burst-release may lead serum levels to exceed the toxic limits, causing localized tissue damage.

**Table 2 pharmaceutics-15-00014-t002:** Summarized table of key findings of revolutionizing therapeutic delivery with microneedle technology for tumor treatment.

	Microneedles-Assisted Therapies	Active Moiety	Key Findings
**Chemotherapy**			
1	MNs	Lipid-coated cisplatin nanoparticles (LCC-NPs)	80% high drug loading achieved. In vivo study on xenograft tumor animal model showed that MNs-mediated delivery had an apoptotic index of 58.6% and reduced tumor volume and weight. Only 20% tumor proliferation in the MNs-based formulation group, whereas it was 60% in the group that received the formulation systemically [21].
2	Miniaturized porous silicon (p-Si) bioresorbable MNs	Doxorubicin	Significant inhibition of melanoma with DOX-loaded p-Si needles, whereas by syringe the same dose failed to inhibit the growth of the tumor. For ~24 h, more than 80% of the covalently linked DOX was released, longer than control specimens with physically bonded DOX (~8 h) and conventional polymeric MNs (15 min^−2^ h) [22].
**Photothermal therapy**			
1	DMNs fabricated from graphene and poly acrylic acid	Model drug Rhodamine B (RhB)	MNs showed sound photothermal effects under NIR and could also deliver the drug directly to tumor tissues and generate heat simultaneously [24].
2	DMNs system for treatment of malignant tumor melanoma in skin	AIEgen (NIR950)	In an in vivo study on sub-cutaneous B16 tumor-bearing C57 mice, the temperature at the tumor site of the MNs-treated group promptly increased compared to the IV injection group at low doses [25].
3	Two-dimensional (2D) MXene (nanosheets of Nb_2_C) and dissolvable polyvinylpyrroli-done (PVP) MNs		Increase in temperature of about 70 °C and thermal ablation with the aid of NIR laser irradiation (1064 nm). After two weeks, tumor growth suppression was observed to be higher than the control group and an improvement was seen in the survival rate of the mice [26].
**Photodynamic therapy**			
1	Sodium hyaluronate (HA) fabricated MNs patch	5-Aminolevulinic acid as photosensitizer	ALA HA MNs with a low dose of ALA (0.61 mg) showed a 97% tumor inhibition rate compared to ALA injection with a high dose of ALA (1.65 mg). The recovery of microchannels due to the MNs patch was quicker (within 3 h) [27].
2	MNs formed of mucoadhesive copolymer Gantrez^®^ AN- 139	Hydrophobic dye NR loaded PLGA NPs	Tissue penetration studies were conducted on excised porcine skin, which showed a high dye concentration (382.63 ng cm^−3^) in the tissue [28].
**Immunotherapy**			
1	DMN	Hepatitis B core protein virus-like particles (OVA-HBc VLPs) fused with the tumor antigen peptide. Mesoporous silica nanoparticles (MSN) as adjuvant.	In vitro, MSN/OVA-HBc@MNs dramatically promoted DC maturation and enhanced OVA presentation on DCs [29].
2	DMNs cocktails	Bioresorbable polypeptide matrix with nanopolyplex, comprising cationic amphiphilic conjugates with ovalbumin-expressing plasmid OVA (pOVA) and immunostimulant-polyinosinic: polycytidylic acid (poly(I:C))	pOVA and poly(I:C) were effectively transferred into the intracellular compartments of macrophages and dendritic cells. A study conducted on 6-week-old BALB/c female mice showed a more robust antigen-specific antibody response than traditional vaccines [30].
3	MNs fabricated with water-soluble polymers	Pluronic F127 (F127) NPs loaded with R837 and coated with cancer cell membranes	MNs inhibited cancer cell growth during prophylactic vaccine therapy and immunotherapy. After immunization using MNs on day 21, compared to the control group, the tumor volume retarded 8-fold, demonstrating tumor growth suppression [31].
4	Biodegradable MNs patch	Cytotoxic T-cell epitope peptide conjugated to HA	Exhibited an effective response for more than 24 h post administration and demonstrated statistically significant tumor growth inhibition in a B16 melanoma mice model [32].
**Gene Therapy**			
1	DMNs	PEI/siRNA	Compared with the untreated animal, the tumor volume and weight declined by ~80% after treatment with a dose of 264 µg of STAT3 siRNA, while the tumor weight and volume reduced by ~50% after treatment with a dose of 132 µg of STAT3 siRNA [33].
2	Coated MNs	siBraf delivery system based upon cell-penetrating peptide octaarginine (R8) nano-complexes	A study conducted on a murine melanoma model induced in Balb/c-nude mice showed ~31% downregulation of BRAF mRNA by R8/siBraf nanocomplexes in vivo, and significant downregulation of BRAF protein expression by coated MNs in vivo [34].
	MNs	NP-loaded RALA (a cell-penetrating cationic peptide) and pDNA	The efficacy of this formulation was checked in a preclinical cervical cancer model, where it evoked a 1.7-fold higher antigen-specific humoral response than those elicited by intramuscular administration [35].
**Combination therapies**			
1	Chemo + PTT MNs consisting of PVP with chitosan for cervical cancer therapy	Doxorubicin and gold-core silica shell (AuMSS) nanorods	These MNs patches showed good photothermal properties, increasing by 12 °C under NIR light irradiation (808 nm, 1.7 W/cm^−2^ for 5 min) [37].
2	Chemo + PTT MNs	Cu-PDA NPs	On the B16F10 tumor-bearing mouse, the obtained core-shell MNs showed excellent antitumor activities in vivo [38].
3	Chemo + PDT MNs of vinylpyrrolidone-vinyl acetate copolymer (PVPVA) for breast cancer therapy	Cisplatin and photosensitizer IR820	MNs-mediated cisplatin delivery (chemotherapy) and IR820 delivery (PDT) alone showed only a moderate ratio, i.e., 52.0% and 64.0%, of tumor inhibition, respectively, whereas MNs-mediated combination delivery showed a higher inhibition ratio of the tumor of 90.0% [39].
4	Immuno + Chemo MNs patch	Anti-PD-1 (aPD-1) and cisplatin-loaded lipid NPs	An aPD-1/CDDP@NPs caused cell apoptosis of 24.5%, however 2.79% cell apoptosis was caused by free CDDP. The cell apoptotic index was 61.4% in the aPD-1 MNs group and 73.2% in the aPD-1/CDDP@NP MNs group, which was considerably greater than 5.4% in the aPD-1 group [40].
5	Gene therapy + PTT MNs patch fabricated with HA	p53 DNA and IR820 as photosensitizer	In in vitro antitumor efficacy studied in a BALB/c nude mice model with a subcutaneous tumor, combination therapy showed the tumor volume was only 40% of the initial volume, which was lowest compared to monotherapies [41].
6	PDT + immunotherapy MNs fabricated with HA	Dextran NPs-loaded zinc phthalocyanine and aCTLA4 antibodies	An in vivo antitumor immune response studied on female BALB/c mice demonstrated that the cytotoxic T-lymphocyte count and T lymphocytes count in the group receiving combination therapy was higher (7.58% and 7.04%) than the group receiving only immunotherapy (0.27% and 0.65%) and only PDT (1.32% and 1.28%) and the control group (0.90% and 1.05%) [42].
**Diagnosis- as biosensors**			
1	MNs sensor on the endomicroscope for NO sensing MNs of polycaprolactone (PCL) and poly (3,4-ethylene dioxythiophene (PEDOT) coating	Hemin molecules for targeting	On testing the microsensor on a mouse skin melanoma model, the NO levels observed in melanoma tissue were two times higher than in normal tissue. It can also identify polyp regions in the colon [43].
2	VEGF sensor with MNs		The biosensor measured both small and large variations in capacitance with a 15 fMrms resolution in the 0.1 to 1000 pM range, therefore, showing a high selectivity of VEGF detection [44].
3	Tyrosine kinase (TYR) sensor wearable bandage device incorporating MNs biosensors		The obtained signal from the current was proportional to the level of TYR present [45].
**Diagnosis- as sampling tool**			
1	Swellable MNs patch made from methacrylated hyaluronic acid (MeHA)		MNs extracted 1.4 ± 0.3 mg (≈1.4 ± 0.3 µL) of ISF in 1 min and 2.3 ± 0.4 mg (≈2.3 ± 0.4 µL) of ISF in 10 min from an experiment conducted on mice [46].
	Swellable hydrogel MNs with gelatin methacryloyl (GelMA)		In an in vivo study on rats, the patch effectively and quantitatively detected glucose and vancomycin in the extracted ISF [47].
2	MNs fabricated with an alginate−peptide nucleic acid (PNA) hybrid material		These MNs patches had the capacity of collecting samples of up to 6.5 μL of fluid in 2 min, hence, providing faster sampling kinetics and minimally invasive sample collection [48].
**Theranostics**			
1	MNs-assisted deep brain imaging and in vivo laser ablation to diagnose and treat brain tumors		In the study conducted on a patient-derived mouse model for the imaging of deep brain cancer, the device demonstrated its potential to detect in vivo lesions in the deep brain with a 1.23 mm imaging depth and 2.4 µm axial resolution in air; minimal invasiveness; quantitative tissue attenuation property (ability to differentiate brain cancer from non-cancer tissues); and high speed. It also performed the function of tissue ablation with an ablation depth of about 600 µm [49].
2	MNs on bioelectronics system for brain tumor diagnosis and treatment		It was observed that the negative control group showed a 520% change in mean tumor volume, and the therapy group showed a −12% change after two weeks. This affected the survival rate where the therapy group had a 42-day median survival time, which was longer than the negative control group by 24 days [50].
	1. HA-based bioresorbable microneedles (BMNs)	Theranostic NPs	
	2. Epoxy-based light-guiding microneedles (LMNs)	Stretchable bioelectronic system	
3	Hollow MNs devices	Quantum dots (QD)	By using MNs, the QD solution was successfully delivered to the dermis and deep epidermis. This enabled the simultaneous multiphoton microscopic imaging of the QD and drug delivery, thus, providing theranostic treatment [51].

**Table 3 pharmaceutics-15-00014-t003:** Clinical trials for microneedles in tumor therapy.

NCT Identifier No.	Study Title	Samples Studied	Conditions	Status
NCT04928222	Placebo MNs in Healthy Volunteers (Part I) and Efficacy/Safety of DOX MNs in Basal Cell Cancer Subjects (Part II) [51]	MNs with placebo and Dox-loaded MNA	Basal Cell Carcinoma	Recruiting
NCT02192021	MNs Array-DOX (MNA-D) in Patients with Cutaneous T-cell Lymphoma (CTCL) [53]	Dox-loaded MNA	Cutaneous T Cell Lymphoma	Recruiting
NCT03646188	Open-Label Dose Escalation Trial to Evaluate Dose-Limiting Toxicity and Maximum Tolerated Dose of MNs Arrays Containing Doxorubicin (D-MNA) in BCC [54]	MNs with placebo and 25 µg, 50 µg, 100 µg, 200 µg of DOX-containing MNA	Basal Cell Carcinoma	Completed
NCT01812837	The Use of MNs in PDT [55]	MNs-loaded Aminolevulinic Acid Radiation: Blue light	Actinic Keratosis	Completed
NCT02594644	The use of MNs to expedite treatment Time in PDT [56]	MNs-loaded Aminolevulinic Acid Radiation: Blue Light	Keratosis, Actinic	Completed
NCT02632110	MNs Lesion Preparation Prior to Aminolevulinic Acid PDT (ALA-PDT) for AK on Face [57]	ALA topical Solution vehicle and MN lesion preparation	Actinic Keratosis	Completed

**Table 4 pharmaceutics-15-00014-t004:** Summarized table of patents with respect to manufacture and applications of various microneedles for tumor therapy.

S.N.	Patent No.	Year of Grant	Title	Description
1.	US7976571B2	2011	PDT irradiation System for the treatment of superficial hyperproliferative tissue growth.	The device in this invention is used for tumor treatment by PDT and involves a plate containing hollow MN array, where optical fibers are individually threaded through the MNs. This is followed by the administration of a photosensitizer, either locally or systemically. After allowing sufficient time to diffuse into the treatment vicinity, a laser with a particular wavelength is linked to the fiber bundle, directing the radiation across the optical fibers to the site of treatment [59].
2.	US20160136407A1	2016	MN arrays for cancer therapy applications.	This invention is about tip-loaded MN used for the delivery of anti-neoplastics, vaccines, and nucleic acid constructs, among other drugs. This system includes the incorporation of a drug into the matrix of the MN, which further converts to hydrogel. This technology can also be used for gene therapy [60].
3.	CN109701152A	2019	A kind of soluble MN patch and preparation method of loading drug.	In this invention, a soluble MN adhesive system was discussed, which was utilized to deliver drugs and vaccines, such as biological molecules, monoclonal antibodies, peptides and proteins, nucleic acids, growth factors, hormones, and polysaccharides. This can also be applied for immunological preparations, such as vaccines for rabies, influenza, rotavirus, polio, hepatitis B, and tumors. The fabrication technique is based on the micro-pin molding process. Multiple advantages of this device are that the manufacturing process is simple, and the device is safe, has good solubility and mechanical properties and a high rate of drug loading [61].
4.	WO2019139827A1	2019	Wearable device with MN array delivery system.	This invention is about a wearable device used to move and support MNs. This device comprises a base connected to a user; a sunken segment molded in the base; an MN array sustained by the base; and an actuator attached to the MN array for its movement in and out of the sunken portion in the base. The MNs are hollow and used to analyze the health status of a patient by the collection of a fluid sample, followed by transmittance to a remote electronic processor for diagnosis and evaluation [62].
5.	WO2021072313A1	2021	Silk fibroin-based MNs and uses thereof.	The invention regards silk fibroin-based MNs tips for the sustained dermal delivery of an anticancer agent and/or an immunomodulatory agent, as well as methods of manufacturing. It also discusses methods and compositions for burst-release or sustained-release, thus, providing an improved immune response [63].
6.	CN108578355B	2021	Transdermal drug delivery MN system with photothermal effect, and preparation method and application thereof.	The invention offers a transdermal drug delivery MN system with preparation techniques, which have a photo-thermal effect due to the presence of embedded graphene nanosheets. It is formed of water-soluble polymer polyacrylic acid. It has multiple advantages, such as high drug loading capacity, improved drug delivery, quick release, and no limit on the drug types. Upon alteration of the content of graphene nanosheets, heating adjustments can be made and, therefore, can accomplish the treatment function with multiple modes [64].
7.	CN104853802B	2021	Method and apparatus for detecting and acquiring biomarkers.	In this invention, probes for one or more biomarkers of interest are attached to an array of MNs. Via skin puncture, the device can be applied directly for biomarker detection in the body through tissue or blood flow [65].
8.	EP3804620A1	2021	Enhanced cancer immunotherapy by MN patch-assisted delivery.	This innovative invention regards a self-degradable MN patch made from biodegradable and biocompatible polymers for the delivery of an immunotherapeutic agent (for example, anti-PD1 antibody) in a sustained and physiologically controllable manner [66].

## Data Availability

Not applicable.

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
