# Peer review of "Revolutionizing Therapeutic Delivery with Microneedle Technology for Tumor Treatment"

_pharmaceutics, 2022, doi:10.3390/pharmaceutics15010014_

Round 1
Reviewer 1 Report
The paper reviews the most recent advances of microneedle technology applied in the management of cancer. Overall, the manuscript is insightful for both experts and non-experts in the field and offers a comprehensive description of the microneedles currently evaluated for cancer therapy and diagnosis. The insightful original figures (Figures 1 – 3) and tables on the different types of microneedles and their most important features are a strongpoint of the present manuscript.
The manuscript can be considered for publication, but a revision of the manuscript is needed as, in the present form, the manuscript lacks conciseness and clarity. In most cases the manuscript contains an overly detailed description of the reported studies, including 8 figures reprinted with permissions from the original papers. A more critical and synthetical approach of the summarized studies is suggested. For conciseness, the authors may consider summarizing the data reviewed in sections 3 through 5 in one or several tables.
Additional observations and suggestions are listed below.
Section 1. Introduction: The authors state “To avoid these issues, chemotherapeutic agents are given by i.v. route as it provides advantages such as higher drug plasma levels, quicker onset of action and prevents systemic side effects”. The authors are kindly asked to verify this statement with respect to the lack of systemic side effects.
Section 3.3, page 10. Please add references for the statement: “PDT with 5-Aminolevulinic acid (ALA) as photosensitizer has been used to treat basal cell carcinoma, actinic keratoses, and squamous cell carcinoma. To prevent severe skin photosensitivity, topical local administration is more favorable”
Section 3.3, page 10. Please add references for the statement: “better substitute is the delivery of NR (a model hydrophobic dye and a pre-formed photosensitizer) that enables the penetration of light into the skin when activated at longer wavelengths. Due to its lipophilic nature and high molecular weight, its topical administration is challenging”
Section 3.5, page 12. Please add references for the statement: “However, the nature of siRNA is hydrophilic and also has a high molecular weight (~13 kDa) with a negative charge which hinders the penetration of these agents through the stratum corneum. Also, the delivery of these agents is required in the basal epidermis and upper dermis of the skin as melanocytes lie in that region”
Section 3.6., page 13. Please correct the OH for the radical form HO· and Cu2+ for Cu2+
Author Response
Respected Reviewer,
We, all the authors are thankful to you for providing us your observations and constructive suggestions for the improvement of our manuscript. We agreed with the comments and modified the manuscript according to the comments. Please see the attachment
Thank you once again.

Reviewer 2 Report
In the review paper by Desai and co-authors, revolutionizing therapeutic delivery with microneedle technology for tumor treatment were introduced. They first reviewed the types, materials utilized in the fabrication, and techniques used for the preparation of microneedles (MNs). In the following sections, they introduced the applications of MNs in tumor diagnosis and treatment, and the case studies related to MNs tumor therapy. Overall, this review is well organized, and literatures included are comprehensive. The topic discussed in the review is inspiring for potential researchers who are trying to enter this research field. I think some in-depth discussion need to be added, and some detail need to be revised. Thus, I suggest the acceptance after revising following points.
1. The authors just simply listed the examples in section 3 and 4 without discussion to highlight why this particular example was chosen. Some points should be given regarding the unique advantages, such as improved loading efficiency, enhanced targeted capability, etc.
2. Need to include tables to summarize section 3 and 4.
3. Some figures are large while some are small. The size or width of figures should be unified.
4. The resolution of figure 6, 9, 10 and 11 should be improved.
5. The format of references need to be adjusted according to the requirement from the journal.
Author Response
Respected Reviewer,
We, all the authors are thankful to you for providing us your observations and constructive suggestions for the improvement of our manuscript. We agreed with the comments and modified the manuscript according to the comments. Please see the response letter as an attachment.
Thank you once again.

Round 2
Reviewer 1 Report
The authors have addressed the issues raised during the first round of the reviewing process and the degree of conciseness has improved to a satisfactory degree. The manuscript can be considered for publication after minor spell check of English language.